# Investigations of the mechanisms of interactions between four non-conventional species with *Saccharomyces cerevisiae* in oenological conditions

Oliver Harlé[1], Judith Legrand[2], Catherine Tesnière[3], Martine Pradal[3], Jean-Roch Mouret[3], Thibault Nidelet[3]*

1 STLO, INRAE, Agrocampus Ouest, Rennes, France, 2 GQE-Le Moulon, INRAE, Univ. Paris-Sud, CNRS, AgroParisTech, Université Paris-Saclay, Gif-sur-Yvette, France, 3 SPO, Univ. Montpellier, INRAE, Montpellier SupAgro, Montpellier, France

* thibault.nidelet@inrae.fr

**Data Availability Statement:** All data files for: Kinetic analysis of yeast-yeast interactions in oenological conditions files are available from the Mendeley database (https://data.mendeley.com/

## Abstract

Fermentation by microorganisms is a key step in the production of traditional food products such as bread, cheese, beer and wine. In these fermentative ecosystems, microorganisms interact in various ways, namely competition, predation, commensalism and mutualism. Traditional wine fermentation is a complex microbial process performed by *Saccharomyces* and non-*Saccharomyces* (NS) yeast species. To better understand the different interactions occurring within wine fermentation, isolated yeast cultures were compared with mixed co-cultures of one reference strain of *S. cerevisiae* with one strain of four NS yeast species (*Metschnikowia pulcherrima*, *M. fructicola*, *Hanseniaspora opuntiae* and *H. uvarum*). In each case, we studied population dynamics, resource consumed and metabolites produced from central carbon metabolism. This phenotyping of competition kinetics allowed us to confirm the main mechanisms of interaction between strains of four NS species. *S. cerevisiae* competed with *H. uvarum* and *H. opuntiae* for resources although both *Hanseniaspora* species were characterized by a strong mortality either in mono or mixed fermentations. *M. pulcherrima* and *M. fructicola* displayed a negative interaction with the *S. cerevisiae* strain tested, with a decrease in viability in co-culture. Overall, this work highlights the importance of measuring specific cell populations in mixed cultures and their metabolite kinetics to understand yeast-yeast interactions. These results are a first step towards ecological engineering and the rational design of optimal multi-species starter consortia using modeling tools. In particular the originality of this paper is for the first times to highlight the joint-effect of different species population dynamics on glycerol production and also to discuss on the putative role of lipid uptake on the limitation of some non-conventional species growth although interaction processes.

datasets/wmhcznvgf4/draft?a=c8b0813a-d27a-45c3-88e0-3f6fe5110130, doi: 10.17632/wmhcznvgf4.1).

**Funding:** The authors received no specific funding for this work.

**Competing interests:** The authors have declared that no competing interests exists.

## 1. Introduction

In natural or anthropized environments, microbial species are part of an ecosystem and interact positively or negatively, forming a complex network. Until recently, process optimization in agriculture or food processing was mostly based on the selection of single strains. However, this paradigm is now being challenged and the scientific community is increasingly seeking to exploit and optimize consortia of several strains and/or species. Indeed, many studies have shown that more diverse anthropized environments have many advantages in terms of resilience, disease resistance or yield. Efforts are now being made to design optimal consortia of various species and strains whose interactions will be exploited to maximize given criteria such as fermentation quality, aromatic complexity or other organoleptic characteristics.

Wine fermentation is both an economically and societally important food ecosystem, where the addition of fermentation 'starters' composed of selected yeasts at the beginning of the fermentation process is common. In fact, around 80% of oenological fermentations worldwide are conducted with starters [1,2]. Most often, these "starters" are only composed of a single *Saccharomyces cerevisiae* (S. c.) strain selected for its ability to complete fermentation. Indeed, numerous experiments have shown that *S. cerevisiae*, with an initially low population, most often becomes the predominant species at the end of the fermentation, demonstrating its superior fermentative abilities [3–6]. However, in recent years, multi-species starters have emerged, aimed at increasing the aromatic complexity of wines. They most often combine one strain of *S. cerevisiae* allowing to complete fermentation and another species, often from a different genus, contributing to a greater variety of flavors [3,4,6,7].

Indeed, there are numerous experiments and even industrial products making use of such mixed starters to improve wine's organoleptic qualities [4,8,9]. The non-*Saccharomyces* (NS) strains used in these experiments are very diverse, with more than 23 different species including *Torulaspora delbrueckii*, *Metschnikowia pulcherrima*, *Metschnikowia fructicola*, *Hanseniaspora opuntiae* and *Hanseniaspora uvarum*. Species in the *Metschnikowia* genus ferment poorly in oenological conditions but can have interesting attributes: in conjunction with *S. cerevisiae*, a strain of *M. pulcherrima* could reduce ethanol concentrations [6,10], increase 'citrus/grape fruit' and 'pear' attributes [11], as well as allow the persistence of 'smoky' and 'flowery' characteristics [12]. *M. pulcherrima* also has an amensalism effect on *S. cerevisiae* through iron depletion via the production of pulcherriminic acid [13]. *M. fructicola* has been less studied and never in conjunction with *S. cerevisiae* although it presents the interesting ability to inhibit *Botrytis* growth [14]. Last, the *Hanseniaspora* genus, studied in sequential or simultaneous fermentation with *S. cerevisiae*, has been shown to increase volatile compound production during winemaking [6]. It notably increased the 'tropical fruit', 'berry', 'floral' and 'nut aroma' characters [15], that were linked to higher concentrations of acetate esters, esters of MCFAs, isoamyl alcohol, 2-phenylethanol and α-terpineol [16].

Despite these various studies, the composition and protocol of inoculation of these multi-strains starters are still very empirical and only based on the input/output balance, without considering the dynamics of the microbial populations or their interactions. This lack of knowledge about yeast-yeast interactions prevents implementing a rational design of multi-strain starters [17]. To address this problem, we decided to focus our study on population dynamics and metabolites produced during oenological fermentations performed in isolated or mixed yeast cultures. Since our goal was not to obtain optimal mixes but to understand the mechanism of microbial interaction, we chose to compare the population dynamics and yields between monocultures of strains from five species (one *S. cerevisiae* and four NS) and four corresponding mixed cultures always including the *S. cerevisiae* strain as reference. We were thus able to identify key microbial interaction mechanisms that are further discussed.

## 2. Results

In this work, we compared in winemaking conditions, the performance of single cell cultures of five different strains from five yeast species (*Saccharomyces cerevisiae*, *Metschnikowia pulcherrima*, *Metschnikowia fructicola*, *Hanseniaspora opuntiae* and *Hanseniaspora uvarum*) and mixed co-cultures combining each of the four NS species with one GFP-labelled *S. cerevisiae* strain representing 10% of the initial inoculate. We chose to stop the monitoring of fermentation at a given time, even if the sugar supply was not completely exhausted. Thus, for all fermentation with the *S. cerevisiae* reference strain, sugars were exhausted after around 200–220 h while in fermentations with single NS strains, the sugar supply was still not exhausted after 400h. Here, we focused on the first 300 hours of fermentation.

By comparing the output of single strain and mixed strain cultures, we evaluated the intensity of yeast-yeast interactions and/or their consequences on ecosystem service production.

### 2.1. $CO_2$ kinetics

We first investigated the influence of species and co-culture on the dynamics of $CO_2$ production (proportional to sugar consumption), which is a good indicator of the fermentation progress. Indeed, $CO_2$ production is easy to monitor (based on weight measurement) and is directly proportional to ethanol synthesis and sugar consumption. The values of the maximum rate of $CO_2$ production (**Vmax**, Fig 1A) and of the maximum $CO_2$ produced were estimated (**$CO_2$max**, Fig 1B). **Vmax** was highly dependent on the species (p.value < 0.001): *S. cerevisiae* cultures (**Sc**) displayed the highest value (**$Vmax_{Sc}$** = 0.99 ± 0.02 g.L$^{-1}$.h$^{-1}$), followed by both

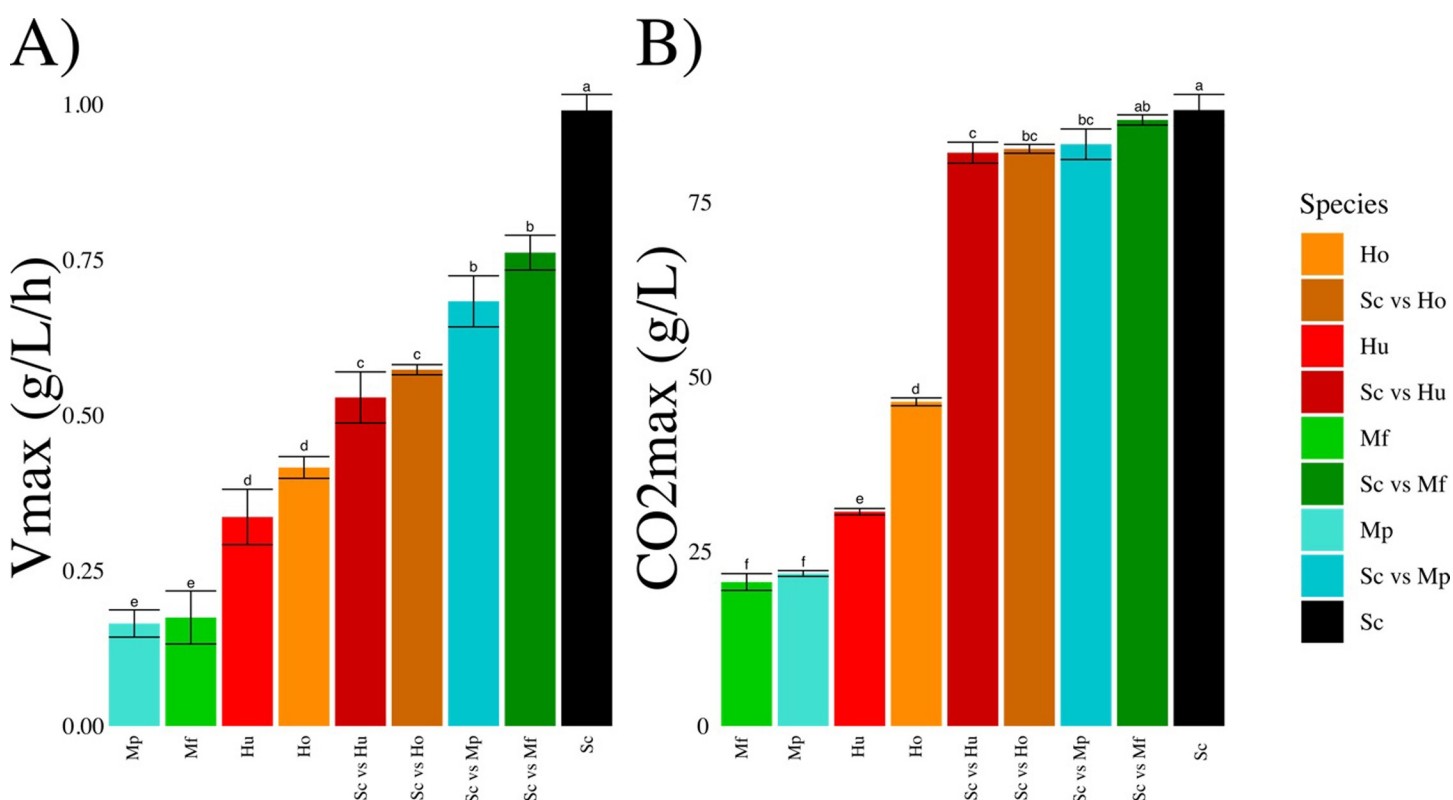

**Fig 1.** Maximum rate of $CO_2$ production, *Vmax* (A) and total $CO_2$ produced (B) in function of the single or mixed species driving each fermentation. Values correspond to average ± standard deviation. The small letters indicate the statistical groups from a Tukey analysis.

*Hanseniaspora* species ($Vmax_{Hu}$ = 0.33 ± 0.04 g.L$^{-1}$.h$^{-1}$, and $Vmax_{Ho}$ = 0.42 ± 0.02 g.L$^{-1}$.h$^{-1}$) and finally both *Metschnikowia* species ($Vmax_{Mp}$ = 0.165 ± 0.02 g.L$^{-1}$.h$^{-1}$ and $Vmax_{Mf}$ = 0.17 ± 0.04 g.L$^{-1}$.h$^{-1}$). The four mixed cultures had intermediate $Vmax$ values between those of *Sc* and the highest $Vmax$ of all NS cultures (Fig 1A). Mixed cultures containing *Metschnikowia* species had significantly higher $Vmax$ values than those containing *Hanseniaspora* species (Fig 1A). Although we did not monitor all cultures until the exhaustion of glucose and fructose, it was however possible to estimate the capacity of a given species to complete fermentation by estimating the amount of $CO_2$ produced during the first 300 hours. *Sc* fermentations finished after around 220 hours with a $CO2max_{Sc}$ = 88.2 ± 2.2 g.L$^{-1}$, all other fermentations did not complete it within 300 h. After 300 h, all mixed cultures are producing $CO_2$ and produced more than 80g $CO_2$.L$^{-1}$ (90% of *Sc* maximum) while both *Hanseniaspora* ($CO_2max_{Hu}$ = 30 ± 0.4 g.L$^{-1}$, and $CO_2max_{Ho}$ = 46 ± 0.6 g.L$^{-1}$) and *Metschnikowia* ($CO_2max_{Mp}$ = 22 ± 0.4 g.L$^{-1}$ and $CO_2max_{Mf}$ = 20 ± 1 g.L$^{-1}$) monocultures stop producing $CO_2$ at 300 h and will never finish the fermentation.

## 2.2. Population kinetics

We also looked at population dynamics in each culture (Fig 2) and determined the maximum growth rate of the population ($\mu$), the maximum population size, also termed carrying capacity ($K$) and the relative abundance (by cytometry) of each species after 300 hours of mixed culture, corresponding in our case to the end of the monitoring period (S1 Table). Fermentations with *S. cerevisiae* alone went through an exponential growth rate ($\mu_{Sc}$ = 0.15 ± 0.02 h$^{-1}$) and reached a maximum population of around $1.5*10^8$cells.mL$^{-1}$ ($K_{Sc}$ = 1.55 ± 0.15 $10^8$cells.mL$^{-1}$) that remained constant until the end of the fermentation. Fermentations with either *Hanseniaspora* species alone had a growth dynamic like *Sc* at the beginning of the fermentation but a higher growth rate ($\mu_{Ho}$ = 0.19 ± 0.03 h$^{-1}$, $\mu_{Hu}$ = 0.62 ± 0.18 h$^{-1}$). In contrast, their stationary phase was quite different from that of *Sc* and characterized by a higher cell mortality with a population drop of about 70% by the end of the process. Fermentations performed by *Metschnikowia* species in monocultures had growth dynamics mostly similar to *Sc* fermentations: a similar growth rate ($\mu_{Mp}$ = 0.18 ± 0.03 h$^{-1}$, $\mu_{Mf}$ = 0.17 ± 0.2 h$^{-1}$), no mortality during the stationary phase but a much reduced maximum population ($K_{Mp}$ = 0.57 ± 0.01 $10^6$cells.mL$^{-1}$, $K_{Mf}$ = 0.8 ± 0.05 $10^6$cells.mL$^{-1}$). In most cases, mixed cultures displayed an intermediate pattern between the two corresponding monocultures (Fig 2). However, mixed or monocultures with *Metschnikowia* displayed different cell mortality rates during the stationary phase: in the case of *ScvsMp* fermentations, only the *S. cerevisiae* population decreased significantly during the stationary phase, while in *ScvsMf* fermentations, both subpopulations significantly decreased. As a measure of fitness, we also followed the variations of *S. cerevisiae* frequency along the fermentation. In all mixed cultures, *S. cerevisiae* was found dominant (frequency > 50%) in the end, increasing significantly during fermentation from 10% initially to frequencies varying between 50% (*ScvsMp*) and 96% (*ScvsMf*) (Fig 2).

## 2.3. Sugar and nitrogen assimilable source consumption

We then looked at the final concentration of resources: sugars (fructose and glucose) and nitrogen assimilable source (NAS) i.e. ammonium and amino-acids (Fig 3). In *Sc* fermentations, less than 0.1% of the initial concentration of both sugars remained (Fig 3A). As seen in the paragraph concerning $CO_2$ production, NS species in monocultures did not complete fermentation in the 300 h period and left respectively 45% of sugars for *Ho*, 67% for *Hu*, 68% for *Mf* and 71% for *Mp*. Furthermore, all species except *H. opuntiae* preferentially consumed glucose (S1 Fig). Sugar consumption was higher in mixed cultures than in single NS species

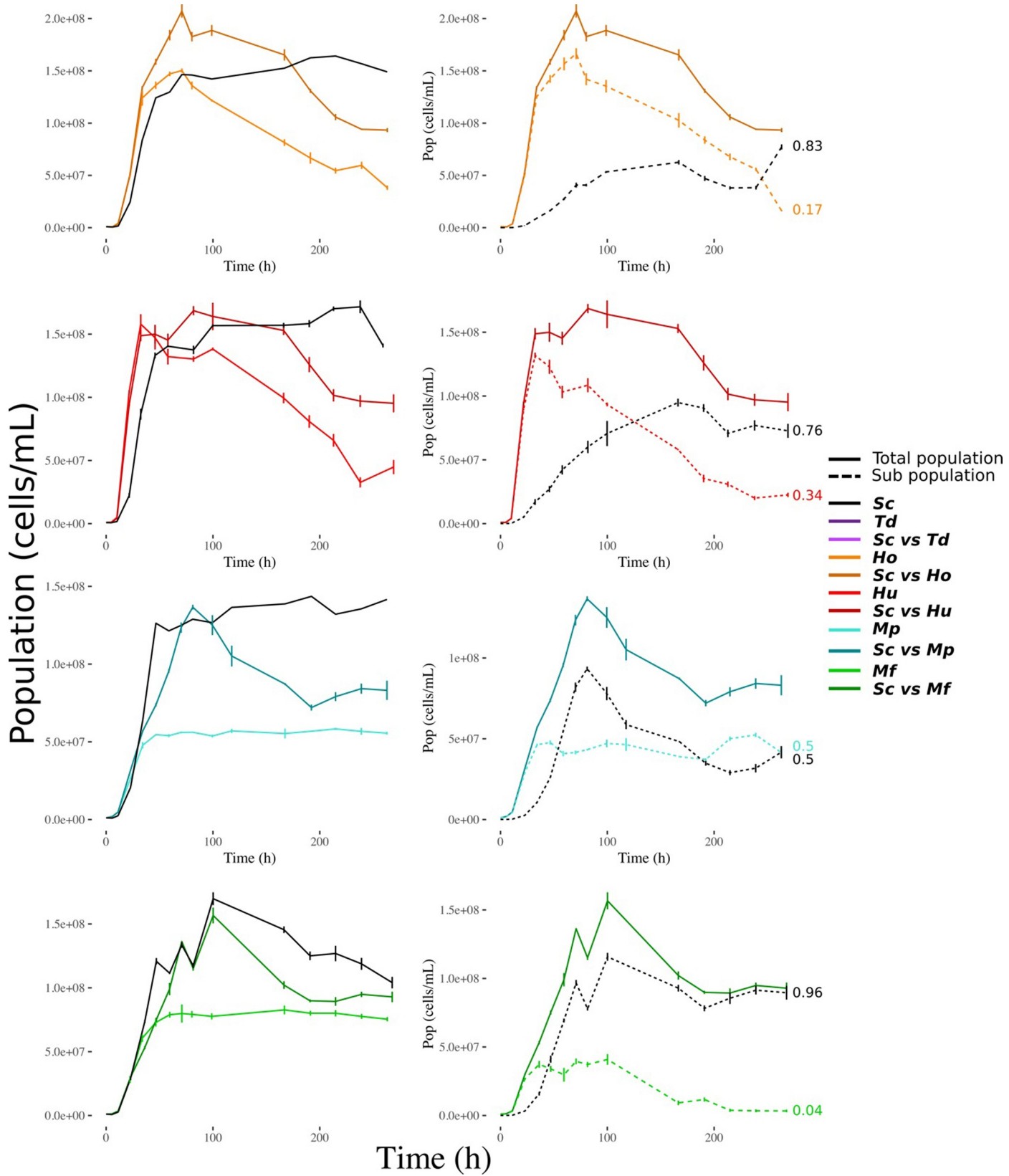

**Fig 2. Global monitoring of the kinetics of the total living population (left), and sub-population in the mixed cultures (right) across fermentation.** Each population was detected by flow cytometry as indicated in the Material and Methods section. Each point represents a sample (average ± standard error). Full lines are for total population and dashed lines for the two sub populations in mixed cultures. At the end of dashed lines, the final proportion of both sub-populations in mixed cultures is indicated. The light colors represent monocultures of 'non-*Saccharomyces*' species and dark ones to the corresponding culture in competition with *S. cerevisiae*. The single strand cultures of *S. cerevisiae* are represented in black.

cultures (Fig 3A). However, it was still lower than in *Sc* species cultures, also with a preference for glucose. This indicates the major impact of *S. cerevisiae* on sugar consumption (consistent with the $CO_2$ production observed), compared to the other NS species studied.

The consumption of NAS displayed the same pattern (Fig 3b). NAS were almost entirely consumed both in *Sc* monocultures and in all co-cultures. In NS monocultures the consumption of NAS varied between 84% and 94%. However, the preference for different nitrogen sources varied with each NS species (Fig 3c). Both *Hanseniaspora* species had similar behaviors, consuming only half of the available ammonium, 90% of histidine and 89% or 79% of arginine (Fig 3C). *Metschnikowia* species presented a similar pattern. It was possible to classify these NS species preferences for the various NAS. The resulting ranking by order of preference was glutamine, methionine, glutamate, valine, threonine, serine, tryptophan, alanine, histidine, arginine, aspartate, glycine and, surprisingly last, ammonium.

## 2.4. Metabolite production

In parallel with must resources consumption monitoring, we also investigated the production of metabolites from Central Carbon Metabolism (CCM): ethanol, glycerol, succinate, pyruvate, acetate and alpha-ketoglutarate (Fig 4). These measurements of metabolite production were taken after 300 hours when sugars consumptions were quite different from one culture to another depending on their dynamics. To allow figures comparison, we computed the production yield (total production / sugar consumption) for each culture and, from these data, we then estimated this yield relatively to that of *Sc* in single strain culture (Fig 4).

In the case of ethanol, only *Mf* fermentations had a relative yield significantly inferior (-32%) to 0 (0 being *Sc* yield). For glycerol, all NS monocultures had a greater yield than *Sc* and mixed fermentations were intermediate between (but not significantly different from) the corresponding monocultures. For acetate, only *Hanseniaspora* species displayed a higher yield (Fig 4).

Finally, all mixed cultures seemed to have a lower succinate yield than both corresponding monocultures (but not significantly after correction for multiple tests).

For each fermentation, the total production of metabolites resulted from the combination of species yields, total sugar consumption and respective population dynamics during fermentation. Therefore, differences observed in the total productions of mixed cultures were the consequences of additive or subtractive effects observed for these 3 components. Considering ethanol, its total production was directly linked to the consumption of resources and all mixed cultures were equivalent to *Sc* fermentations (S2 Fig). The case of glycerol was more interesting. Indeed, even if the average sugar consumption was lower in *ScvsHu* and *ScvsMp* mixed cultures than in single strain *Sc* culture, the total production of glycerol was significantly higher than that of the corresponding monocultures (Glycerol$_{SCvsMp}$ = 6.1±0.1 g.L$^{-1}$, Glycerol$_{Sc}$ = 5.3±0.4 g.L$^{-1}$, Glycerol$_{Mp}$ = 3.7±0.2 g.L$^{-1}$). This resulted from the positive combination of the greater glycerol yield by *Hanseniaspora* and *Metschnikowia* and their population dynamics. For acetate, *Hanseniaspora* species have a higher production in monoculture compared to *S. cerevisiae* and mixed cultures, whereas it was the converse for *Metschnikowia* species. For all other metabolites, the total production of mixed cultures was not significantly different from the corresponding monocultures (S1 Table).

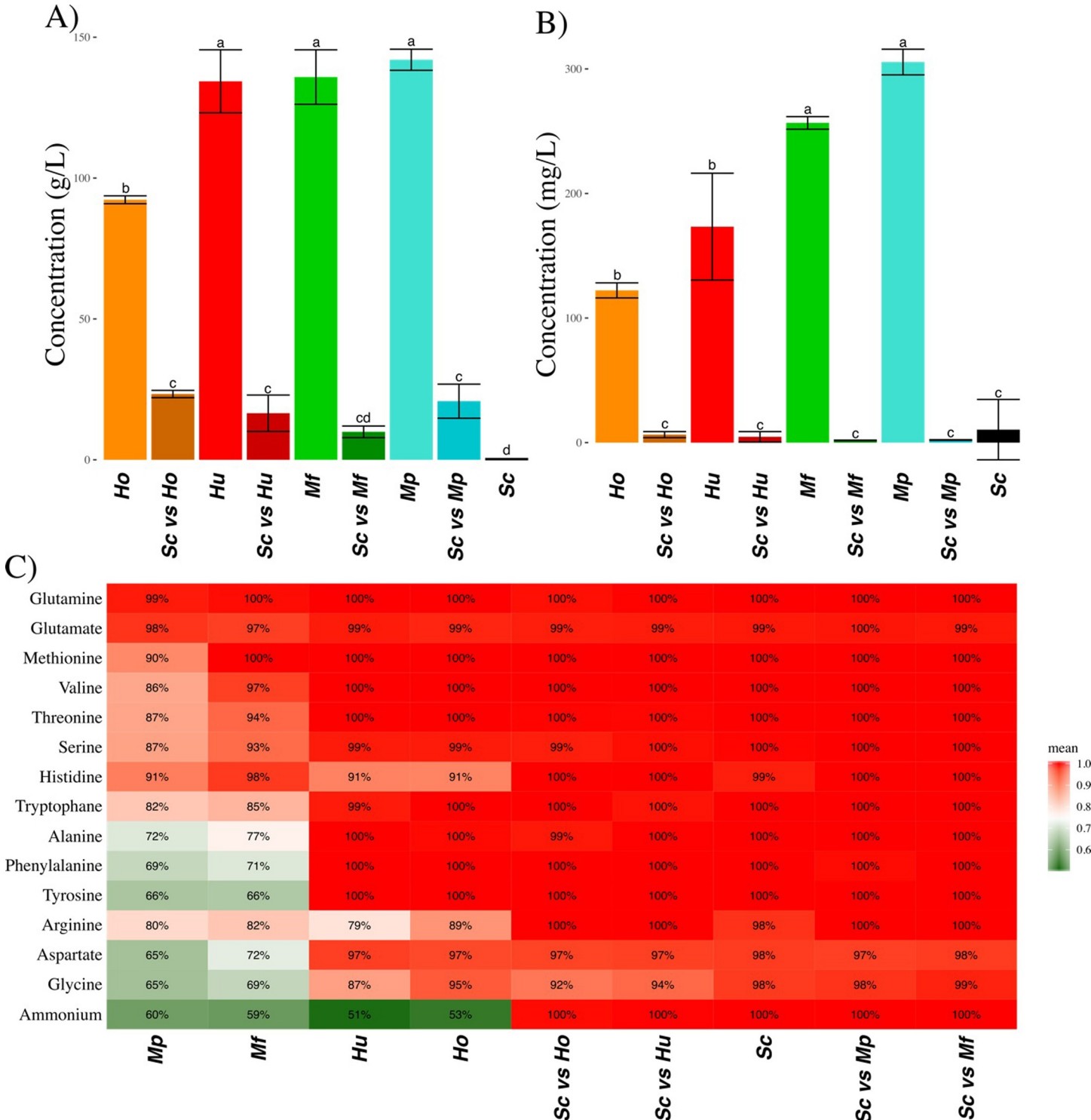

**Fig 3. Consumption of sugars and nitrogen assimilable sources (NAS) for each type of fermentation.** A) Final concentration of sugar (average ± standard deviation). B) Final concentration of NAS (average ± standard deviation). C) Percentage of consumption of each NAS in each type of fermentation represented as a color gradient from green (<75%) to red (> 75%).

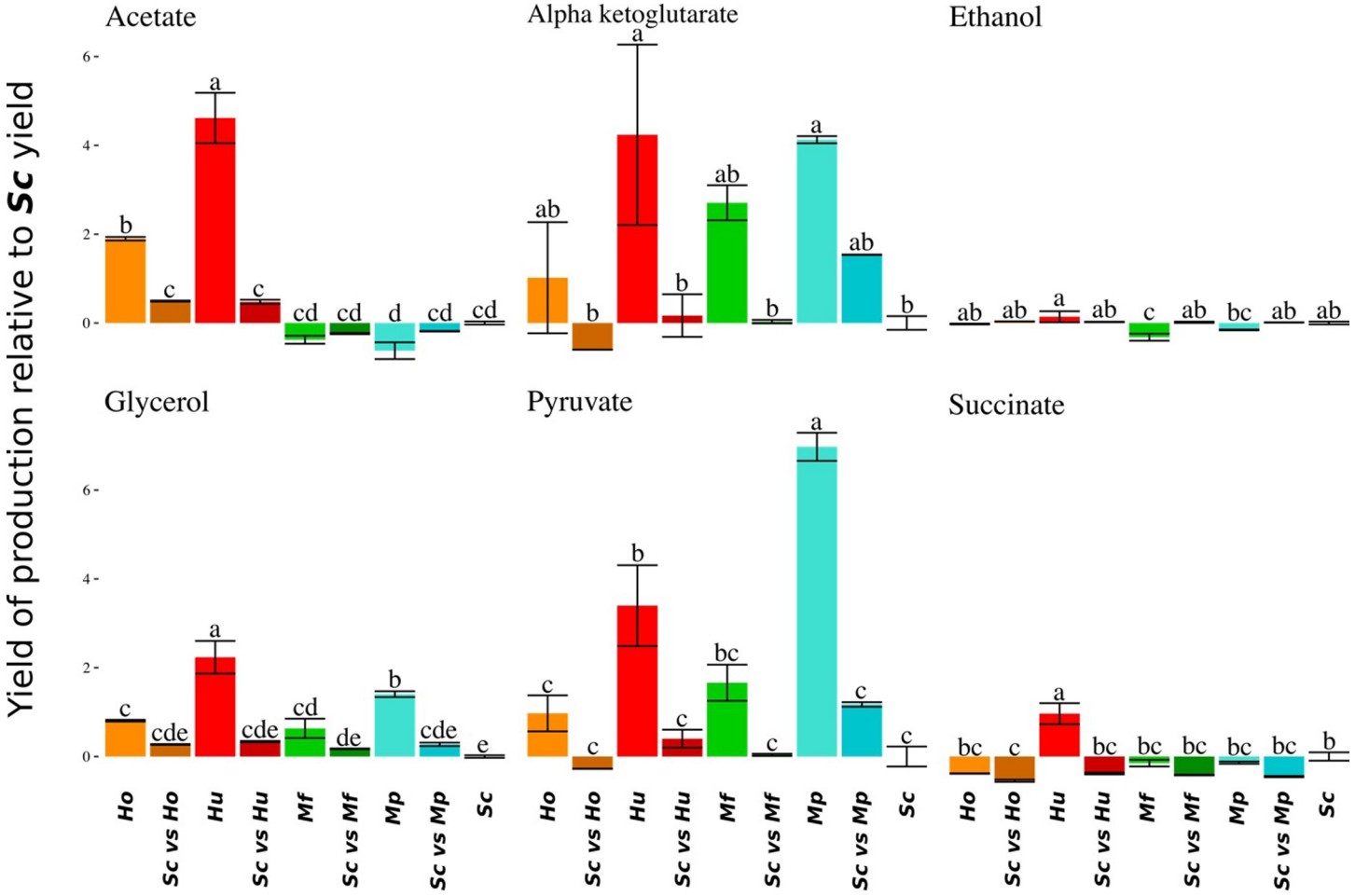

**Fig 4. Yield of carbon metabolite production relative to the yield of production of single *S. cerevisiae*.** Average Yield are given with standard deviations for acetate, alpha-ketoglutarate, ethanol, glycerol, pyruvate and succinate and each type of fermentation.

## 3. Discussion

This study presents one of the first works focusing on the population dynamics and kinetics of yeast-yeast interactions between two species (S.C. and NS) during the alcoholic fermentative process [18]. The counterpart of this deep phenotyping is two limitations: the monitoring time and the number of strains by species. Due to experiment constraints, we followed the fermentations during 300 hours and until sugar exhaustion. Sc fermentations consume all sugars and therefore finish the fermentation in around 200h, while *Metschnikowia* and *Hanseniaspora* monocultures are enabled to finish it (consuming around a quarter of available sugars). However, it is not clear if the mixed cultures are able to finish it. At the end of our experiment all mixed cultures were still producing $CO_2$ and produced more than 80g $CO_2.L^{-1}$ (90% of *Sc* maximum) and still have a high viable population. Although without kinetic tracking, these mixed cultures have already been tested under similar conditions [16,19]. In most of the cases they were able to finish the fermentation. Therefore, it is reasonable to make the hypothesis that all our mixed cultures will finish the fermentation in more than 300 hours. Another limitation is that we only used one strain by species. Indeed, here we studied a limited sample of species and strains that raises the question of the genericity of interactions. This question can be found in other similar papers and is difficult to address as increasing the number of species

or strains increase exponentially the number of mixed cultures to be tested. In our cases we have tested 5 mixed cultures in triplicate. To test all possible mixed cultures (including NS-NS mixed) represent 25 fermentations. With for example 3 strains for each species tested in this study, it would represents 15 strains and 225 mixed fermentations. It is very difficult if not impossible to follow experimentally the kinetics of so much fermentation.

Despite the small number of species and strains, it is important to note that a deep phenotyping was performed in the current work, making it possible a better understanding of the interactions between two yeast strains. Indeed, the following parameters were measured for every experiment: fermentation kinetics, cell growth, cell viability, percentage of non-saccharomyces yeast, concentration of metabolites of central carbon metabolism, nitrogen compounds.

A notable observation is that monocultures studied here can be grouped by genus as respectively *Metschnikowia* and *Hanseniaspora* are more similar between them than across them (S3 Fig). Of course, this observation has to be confirmed with further investigations.

Even though *Metschnikowia* maintains a high viability throughout fermentation, the medium resources (sugars and nitrogen) were not entirely consumed. The reason why the cells stopped growing despite the availability of these resources remains unclear. A possible explanation could be linked to oxygen availability. Our hypothesis is that *Metschnikowia* is not able to import lipid from the extracellular medium and that this species stops growing when oxygen content in the must is equal to zero. For all yeast, lipid synthesis requires oxygen and is therefore impossible in anaerobic conditions [20]. With the progress of fermentation, ensuing oxygen limitation and ethanol accumulation yeast should import lipids to survive [21]. If not, it stops to multiply [22,23]. *Metschnikowia* species growth (studied in this work) depends thus on their initial (internal) lipid content and lipid synthesis from initial concentration of oxygen (see S1 Fig).

Interestingly, even though *S. cerevisiae*, *M. pulcherrima* and *M. fructicola* mortality rates were low in monocultures, the corresponding mixed cultures (***ScvsMp*** and ***ScvsMf***) presented 30% mortality (after 200 hours of fermentation). Moreover, this mortality seemed to impact species differently. In the mixed ***ScvsMp*** culture, only *S. cerevisiae* cells eventually died, whereas in ***ScvsMf*** both species were negatively affected. The survival of *M. pulcherrima* cells compared to *S. cerevisiae* cells could be explained by the production of pulcherriminic acid by *M. pulcherrima* [24]. Indeed, pulcherriminic acid is known to deplete iron from the medium, which has a lethal effect on *S. cerevisiae* cells [13,25,26]. In ***ScvsMf*** cultures, the mortality observed in both species suggests a more complex mechanism of interaction (although it is not clear whether *M. fructicola* also produces pulcherriminic acid [14]. To explain these results, we could hypothesize the conjunction of two different mechanisms of interaction. It could be that *M. fructicola* synthesized a metabolite (pulcherriminic acid?) impacting the viability of *S. cerevisiae* cells (through iron depletion?), with *M. fructicola* cells dying thereafter for another reason such as sensitivity to ethanol. Indeed, the production of ethanol was almost four times higher in mixed cultures than in single strain *Metschnikowia* fermentations. Under such hypothesis, the reason why no loss of viability was observed for *M. pulcherrima* in mixed culture with *S. cerevisiae* could probably be related to a better tolerance of *M. pulcherrima* to ethanol stress compared to *M. fructicola*.

Mixed population dynamics were all characterized by similar growth rate (reaching a maximum population like *S. cerevisiae* single strain culture), followed by a long phase with decreasing viability (NS viability dropped to 30% in accordance with [16,27]). Moreover, their yield and total production of CCM metabolites were very similar. For almost all these characteristics, mixed cultures presented intermediate phenotypes compared to the corresponding monocultures. However, there is a remarkable exception considering the total production of glycerol

that is superior in mixed cultures whereas their sugar consumption was inferior (Fig 4). This point is characteristic of a transgressive interaction (often referred to as over-yielding), i.e. a situation in which the ecosystem performance is higher than the best-yielding species performance in monoculture. This glycerol overproduction in mixed cultures has already been observed in previous works [16,28] but seems to depend on the species and experimental conditions [27]. In the present study, the glycerol and sugar consumption observed in mixed cultures can be explained without any change of individual behavior but by the synergic effects of population dynamics (*S. cerevisiae* slowly dominating the population), resource consumptions (the NS fermentation leaving ⅔ of sugars) and the glycerol yields of NS species that were two to three times higher that of *S. cerevisiae*. These overproductions of glycerol is clear example of how mixing species can produce a better result of any monoculture. Thus exploiting specific yield, population dynamics and inoculation protocol could lead to high-performance fermentations. As overproduction appears to be highly dependent of population dynamics it will be interesting to test different inoculation protocols. Decreasing the frequency of *S. cerevisiae* at t0 (0.01, 0.001, 0.0001) could increase the transgressive interaction and lead to higher glycerol and thus eventually lower ethanol.

So far we only discussed transgressive interactions when mixed cultures over-produced (or under-produced) a given metabolite. Indeed, it was very difficult to identify interactions when the productions of mixed fermentations were within the range of monocultures productions. As the relative frequency of both species in mixed cultures evolved during fermentation, it was difficult to link the final mix to the contributions of each species. It was even more difficult to assess whether these contributions combined additively or with interaction. New statistical developments or dual transcriptomics will be needed to answer this question [29–31].

To discuss more generally the mechanisms of interactions of *Metschnikowia* and *Hanseniaspora* cultures mixed with *S. cerevisiae*, we could not observe any major antagonistic phenomena. For almost all assays, mixed cultures performance stood always between that of the corresponding monocultures (Figs 1, 2 and 3), except for the total production of glycerol. Moreover, despite the differences in yield and interactions between species, the rapid dominance of *S. cerevisiae* (increasing from 10% to at least 50% during the fermentation) resulted in mixed cultures that were overall not different from *S. cerevisiae* monocultures. This result is in agreement with the good adaptation of *S. cerevisiae* to winemaking conditions [4–6,32] but could be different with a different set of species. This result is important in the context of ecological engineering. In fact, our results confirmed that *S. cerevisiae* has a much better fitness than the NS species studied in this paper. Therefore, if we want mixed culture behavior to deviate from that of *S. cerevisiae* single strain culture, it must be ensured that NS cells dominate the culture as soon as possible. To achieve this, two conceivable options are currently tested: either to reduce the proportion of *S. cerevisiae* at $t_0$ or to perform a sequential inoculation [16]: first the NS species and then the *S. cerevisiae* strain in a second time. These two options could be equivalent depending on the type of interaction(s) that occurs. If strain behaviors in single strain or mixed cultures are identical, then all interactions are mediated by the medium through the competition for resources and the production of constitutive toxins such as ethanol (producing a toxin only in mixed fermentation would be a behavior change) and could be qualified as "indirect".

In the case of indirect interaction, mathematical models could be designed from data on monocultures to predict the mixed cultures. This would allow simulating numerous mixes of species with various initial conditions and identify optimal strategies depending on one or several given criteria. Using these approaches could limit the number of necessary tests, potentially saving a lot of time and money and opening the way to a more methodical ecological engineering. The development of such mathematical models will only be possible thanks to a

deep tracking of population dynamics to understand underlying mechanisms of growth and mortality. Obviously, it is also critical to validate this approach by i) first extending the number of species co-cultured with *S. cerevisiae*, ii) investigating intra-specific variability and strain-strain interactions between species, iii) investigating the impact of the environment of culture (temperature, grape variety, nutrient availability, etc.).

## 4. Mat & met

### 4.1. Strains

In this work, we used one strain of 5 different species (one strain per species): *Saccharomyces cerevisiae* (**Sc**), *Metschnikowia pulcherrima* (**Mp**), *M. fructicola* (**Mf**), *Hanseniaspora uvarum* (**Hu**) and *H. opuntiae* (**Ho**). The *S. cerevisiae* strain is a haploid strain from EC1118 labelled with GFP (59A-GFP, [1]). The *Hanseniaspora uvarum* (CLIB 3221) *H. opuntiae* (CLIB 3093) and *Metschnikowia pulcherrima* (CLIB 3235) species originated from the yeast CIRM (https://www6.inrae.fr/cirm_eng/Yeasts/Strain-catalogue) and were isolated from grape musts. The *Metschnikowia fructicola strain was* from the Lallemand collection.

For each strain, 3 replicates of monocultures were performed (except for *S. cerevisiae* that had a total of 8 replicates in different blocks). In addition, for each NS strain, 3 replicates of a mixed culture with the **Sc** strain were performed. In all mixed fermentations, the starting proportion of *S. cerevisiae* cells was set at 10%. In this text, fermentations were referred to by the species that performed them, i.e. monocultures were referred to as: *Sc*, *Mp*, *Mf*, *Hu* and *Ho* and mixed strain cultures as *ScvsMp*, *ScvsMf*, *ScvsHu* and *ScvsHo*.

### 4.2. Medium

Initial cultures (12 h, in 50 ml YPD medium, 28°C) were used to inoculate fermentation media at a total density of $10^6$ cells/mL; therefore, for mixed culture the *S. cerevisiae* cells density was $0.1 \times 10^6$ /mL and the NS cells density was $0.9 \times 10^6$ /mL. Fermentations were carried out in a synthetic medium (SM) mimicking standard grape juice [33]. The SM used in this study contained 200 g/L of sugar (100 g glucose and 100 g fructose per liter) and 200 mg/L of assimilable nitrogen (as a mix of ammonium chloride and amino acids). The concentrations of weak acids, salts and vitamins were identical to those described by [34]. The pH of the medium was adjusted to 3.3 with 10M NaOH. The SM medium was first saturated with bubbling air for 40 minutes, then it was supplemented with 5 mg/L phytosterols (85451, Sigma Aldrich) solubilized in Tween 80 to fulfill the lipid requirements (sterols and fatty acids) of yeast cells during anaerobic growth.

### 4.3. Measurements

Fermentation took place in 1.1-liter fermentors equipped with fermentation locks to maintain anaerobiosis, at 20°C, with continuous magnetic stirring (500 rpm) during approximately 300h. $CO_2$ release was followed by automatic measurements of fermentor weight loss every 20 min. The amount of $CO_2$ released allowed us to monitor the progress of the fermentation and evaluate the maximum of released $CO_2$ (***CO2max***) as well as the maximum rate of $CO_2$ released (***Vmax***). Samples were harvested after 6h, 12h and 24h, then every 12h during the first week and every 24h during the second week of fermentation. For each sample, the population density cells were determined using a BD Accuri™ C6 Plus flow cytometer as described in [35]. Viability was determined using propidium iodide staining and BD Accuri™ C6 Plus flow cytometer adapted from [36]. Proportions of *S. cerevisiae* in mixed-culture was established thanks to the green fluorescence produced by the *S. cerevisiae* 59A-GFP, the forward and side

scatters from the BD Accuri™ C6 Plus flow cytometer and machine learning using the caret package in R [37]. From these population densities (without taking into account viability), we fitted a growth population model (with the growthcurver package in R, [38], and determined the carrying capacity (**K**) and maximum growth rate (**mu**) for each fermentation.

The final concentrations of carbon metabolites in the medium (acetate, succinate, glycerol, alpha-ketoglutarate, pyruvate, ethanol, glucose and fructose) were determined with high-pressure liquid chromatography [39]. From these metabolite concentrations, we first calculated the consumed sugar concentration as the difference between the final and the initial concentration of either glucose or fructose. Then we calculated the yield of metabolite production by dividing the final concentration by the corresponding consumed sugar concentration. Finally, we compared these yields to the yield of *S. cerevisiae* monocultures considered as reference.

Finally, the ammonium concentration after 100h of fermentation was determined enzymatically with R-Biopharm (Darmstadt, Germany) and the free amino acid content of the must was estimated through cation exchange chromatography with post-column ninhydrin derivatization [40].

### 4.4. Statistical analysis

The experimental work was performed in 5 different blocks. Each block was composed of three replicates of NS fermentations (for example **Hu**), three replicates of the corresponding mixed fermentations with *S. cerevisiae* (for example **ScvsHu**) and one or two fermentations of single strain *S. cerevisiae* cells (**Sc**). The block effect was evaluated on the parameters of the **Sc** fermentation. For most studied parameters, the block effect was not significant. For those parameters where a block effect was observed (**mu** and **K**), a statistical correction for block effect did not modify our results. Therefore, for simplification purposes, we compared all fermentations without any correction for the block effect parameter. For each measured parameter, an ANOVA was performed to evaluate the type of fermentation (**Sc**, **Mp**, **Mf**, **Hu**, **Ho**, **ScvsTd**, **ScvsMp**, **ScvsMf**, **ScvsHu** and **ScvsHo**) effect and then a Tukey t-test was performed to determine statistical groups and two-by-two statistical differences. All statistical analyses were performed using R [41] and Rstudio [42]. All data, analysis and figures scripts can be found in this github address: https://github.com/tnidelet/Git-Harle-et-al-2019.

### Supporting information

**S1 Fig. Glucose and Fructose consumption kinetics in function of different strains.** Each point represents a sample (average ± standard deviation).
(TIF)

**S2 Fig. Total production of carbon metabolite in function of the strains driving the fermentation.** Average production are given with standard deviations for acetate, alpha-ketoglutarate, ethanol, glycerol, pyruvate and succinate.
(TIF)

**S3 Fig. Principal component analysis of carbon metabolites and growth parameter of monocultures.** The mixed cultures are a second time projected on the plan determiner by only monocultures. In the top right is represented the circle of variables.
(TIF)

**S4 Fig.** Serial tenfold dilutions of two Metschnikovia pulcherrima strains (A and B) spotted onto various synthetic standard agar media (SM425, 425 mg/l assimilable nitrogen) with Tween 80, (Tw, 0.06%), supplemented or not with phytosterol (Phyto, 20 mg/L), in the presence or not of fluconazole (FLC, 256 µg/mL). Plates were incubated at 28˚C for five days in air

or in anaerobiosis
(TIF)

**S1 Table. Growth parameter values for each type of fermentation.**
(DOCX)

## Acknowledgments

We thank the CIRM, the Lallemand company, Jean-Luc Legras, Virginie Galeote and Jean-Nicolas Jasmin for providing the species used in this study. We thank also Christian Picou, Marc Perez, Faiza Macna for technical assistance and Delphine Sicard for advices.

## Author Contributions

**Conceptualization:** Jean-Roch Mouret, Thibault Nidelet.

**Data curation:** Oliver Harlé, Judith Legrand, Thibault Nidelet.

**Formal analysis:** Oliver Harlé, Judith Legrand, Catherine Tesnière, Thibault Nidelet.

**Funding acquisition:** Thibault Nidelet.

**Investigation:** Oliver Harlé, Catherine Tesnière, Martine Pradal, Thibault Nidelet.

**Methodology:** Oliver Harlé, Judith Legrand, Catherine Tesnière, Jean-Roch Mouret, Thibault Nidelet.

**Project administration:** Thibault Nidelet.

**Resources:** Catherine Tesnière, Martine Pradal, Jean-Roch Mouret, Thibault Nidelet.

**Software:** Oliver Harlé, Judith Legrand, Thibault Nidelet.

**Supervision:** Thibault Nidelet.

**Validation:** Oliver Harlé, Judith Legrand, Catherine Tesnière, Martine Pradal, Jean-Roch Mouret, Thibault Nidelet.

**Visualization:** Oliver Harlé, Judith Legrand, Catherine Tesnière, Jean-Roch Mouret, Thibault Nidelet.

**Writing – original draft:** Oliver Harlé, Judith Legrand, Catherine Tesnière, Martine Pradal, Jean-Roch Mouret, Thibault Nidelet.

**Writing – review & editing:** Oliver Harlé, Judith Legrand, Catherine Tesnière, Jean-Roch Mouret, Thibault Nidelet.

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
