## [Decision Letter · Decision Letter 0]

10 Feb 2020

PONE-D-19-31984

Investigations of the mechanisms of interactions between four non-conventional species with Saccharomyces cerevisiae in oenological conditions.

PLOS ONE

Dear Dr Nidelet,

Thank you for submitting your manuscript to PLOS ONE. First of all, we are sorry that it took long time to review your manuscript. After careful consideration, we feel that it has merit but does not fully meet PLOS ONE’s publication criteria as it currently stands. Therefore, we invite you to submit a revised version of the manuscript that addresses the points raised during the review process.

Especially, the reviewer #1 raised several concerns. Please address these points one by one.

We would appreciate receiving your revised manuscript by Mar 26 2020 11:59PM. To enhance the reproducibility of your results, we recommend that if applicable you deposit your laboratory protocols in protocols.io, where a protocol can be assigned its own identifier (DOI) such that it can be cited independently in the future. For instructions see: http://journals.plos.org/plosone/s/submission-guidelines#loc-laboratory-protocols

We look forward to receiving your revised manuscript.

Kind regards,

Yoshikazu Ohya, PhD

Academic Editor

PLOS ONE

Reviewers' comments:

Reviewer's Responses to Questions

**Comments to the Author**

1. Is the manuscript technically sound, and do the data support the conclusions?

Reviewer #1: No

Reviewer #2: Yes

2. Has the statistical analysis been performed appropriately and rigorously? 

Reviewer #1: Yes

Reviewer #2: Yes

3. Have the authors made all data underlying the findings in their manuscript fully available?

Reviewer #1: Yes

Reviewer #2: Yes

4. Is the manuscript presented in an intelligible fashion and written in standard English?

Reviewer #1: No

Reviewer #2: Yes

5. Review Comments to the Author

Reviewer #1: The authors made the following assumption : « all cultures that produced more than 80g CO2.L-1 (90% of Sc maximum) within 300 hours will be able to complete fermentation »

This assumption is not supported by any scientific evidence. The authors should have followed the entire fermentation kinetics in order to interpret the data correctly.

In the summary the authors stated : » M. pulcherrima and M. fructicola displayed a negative interaction with the S. cerevisiae strain tested, with a decrease in viability in co-culture, probably due to iron depletion via the production of pulcherriminic acid.

The authors suggest iron depletion to be responsible for viability decrease, however, no results presented in this paper support this hypothesis, already reported.

Summary is not the place for such assumption.

The first part of the discussion is redundant with the result.

One of the conclusion is : Meanwhile NS fermentations present similarities, it was possible to group the NS strains performance according to their genus.

This is an overinterpretation, taking into account the small number of tested genus and tested strains.

The overall discussion is quite poor with very few references to support the comments.

For example, regarding the role of lipids that could play a role in the interactions, there is no reference, while some works do exist.

CO2 should be written CO2 in legend

Reviewer #2: The manuscript is concise and easy to follow. The authors have evaluated the interactions between Saccharomyces cerevisiae and four non-Saccharomyces species with focus on metabolic interactions in particular nutrient resource utilization. A couple of different interaction mechanisms could be identified. One limitation to this study is that only single strains were used for each species, and the authors should at least acknowledge the fact that there might be strain variability and briefly discuss this. Overall, the study set-up is sound but would have benefited from multiple strains per species.

Some minor corrections in text are necessary to improve the quality of the paper. These are as follows:

Throughout the document, replace “isolated cultures” with either ‘single cultures’ or ‘monocultures’

The use of “strain” in this manuscript is confusing since only single strains were used for each species. Perhaps the authors should rather consider just referring to the species.

All the images used for the figures are poor quality and no easy to read through

Page 2 line 8: “and” should not be in italics

Page 3 line 6: “aiming” should be aimed

Page 4 line 75: “smocky” should be smoky

Page 5 line 101: “choose” should be chose

Page 6 line 135: “corresponds” should be correspond

Page 7 line 147: change “on the opposite” to Conversely or In contrast

Page 7 line 151: “µm” should be µM

Page 8 line 178: insert space between 300 and h

Page 8 line 185-186: the part of the sentence starting with “whereas” does not make sense

Page 9 line 217: change “isolated” to monoculture

Page 9-10 line 217-219: the author indicates no significant difference between mixed cultures and their corresponding mono cultures, but this is not true for the Hu and Mp cultures.

Page 11 line 252: “decreasing” is in a different font type from the rest of the text

Page 12 line 275: which PCA and Figure 5 is the author referring to?

Page 12 line 280: remove “nevertheless” and start the sentence with “Despite”

Page 12 line 286 – 290: the authors should refer to studies that have been done on oxygen e.g. Shekhawat et al. 2017 and 2018, Morales et al., 2015

Page 14 line 320: replace “evidence” with observe

6. PLOS authors have the option to publish the peer review history of their article (what does this mean?). If published, this will include your full peer review and any attached files.

Reviewer #1: No

Reviewer #2: No

---

## [Author Response · Author response to Decision Letter 0]

31 Mar 2020

Reviewer #1: The authors made the following assumption : « all cultures that produced more than 80g CO2.L-1 (90% of Sc maximum) within 300 hours will be able to complete fermentation »

This assumption is not supported by any scientific evidence. The authors should have followed the entire fermentation kinetics in order to interpret the data correctly.

This part was not clearly described, thus the sentence has been completely rewrote to clarify the point and data concerning the maximum level of CO2 produced have been given. See lines 125-130. Moreover, this point has been discussed in the discussion section (see lines 251-264 with reference added).

In the summary the authors stated : » M. pulcherrima and M. fructicola displayed a negative interaction with the S. cerevisiae strain tested, with a decrease in viability in co-culture, probably due to iron depletion via the production of pulcherriminic acid.

The authors suggest iron depletion to be responsible for viability decrease, however, no results presented in this paper support this hypothesis, already reported. 

Summary is not the place for such assumption. 

The sentence “probably due to iron depletion via the production of pulcherriminic acid” has been removed from the abstract (see lines 35-36).

The first part of the discussion is redundant with the result. 

This part has been rewrote to take this remark into account (see lines 244-365 with reference added).

One of the conclusion is : Meanwhile NS fermentations present similarities, it was possible to group the NS strains performance according to their genus.

This is an overinterpretation, taking into account the small number of tested genus and tested strains. This point has been discussed lines 257-270.

The overall discussion is quite poor with very few references to support the comments. Numerous changes have been performed in particular with the addition of new references to support our comments (lines 244-365).

For example, regarding the role of lipids that could play a role in the interactions, there is no reference, while some works do exist. References have been added (see paragraph 285-302).

CO2 should be written CO2 in legend (done)

Reviewer #2: The manuscript is concise and easy to follow. The authors have evaluated the interactions between Saccharomyces cerevisiae and four non-Saccharomyces species with focus on metabolic interactions in particular nutrient resource utilization. A couple of different interaction mechanisms could be identified. One limitation to this study is that only single strains were used for each species, and the authors should at least acknowledge the fact that there might be strain variability and briefly discuss this. Overall, the study set-up is sound but would have benefited from multiple strains per species.

Some minor corrections in text are necessary to improve the quality of the paper. These are as follows:

Throughout the document, replace “isolated cultures” with either ‘single cultures’ or ‘monocultures’ “Isolated cultures” has been replaced by “Monocultures” lines 94, 161 and throughout the paper.

The use of “strain” in this manuscript is confusing since only single strains were used for each species. Perhaps the authors should rather consider just referring to the species. This has been done throughout the paper.

All the images used for the figures are poor quality and no easy to read through

Page 2 line 8: “and” should not be in italics. (This has been done line 28.)

Page 3 line 6: “aiming” should be aimed (done line 64)

Page 4 line 75: “smocky” should be smoky (done line 74)

Page 5 line 101: “choose” should be chose (done line 100)

Page 6 line 135: “corresponds” should be correspond (done line 134)

Page 7 line 147: change “on the opposite” to Conversely or In contrast (done lines 147)

Page 7 line 151: “µm” should be µM (done line 151)

Page 8 line 178: insert space between 300 and h (done line 179)

Page 8 line 185-186: the part of the sentence starting with “whereas” does not make sense (sentence changed lines 187-188)

Page 9 line 217: change “isolated” to monoculture (done line 210)

Page 9-10 line 217-219: the author indicates no significant difference between mixed cultures and their corresponding mono cultures, but this is not true for the Hu and Mp cultures. (This has been clarified lines 218-220)

Page 11 line 252: “decreasing” is in a different font type from the rest of the text OK

Page 12 line 275: which PCA and Figure 5 is the author referring to? The paragraph has been removed.

Page 12 line 280: remove “nevertheless” and start the sentence with “Despite” The paragraph has been removed.

Page 12 line 286 – 290: the authors should refer to studies that have been done on oxygen e.g. Shekhawat et al. 2017 and 2018, Morales et al., 2015 (references added line 297)

Page 14 line 320: replace “evidence” with observe (OK)

---

## [Decision Letter · Decision Letter 1]

4 May 2020

Investigations of the mechanisms of interactions between four non-conventional species with Saccharomyces cerevisiae in oenological conditions.

PONE-D-19-31984R1

Dear Dr. Nidelet,

We are pleased to inform you that your manuscript has been judged scientifically suitable for publication and will be formally accepted for publication once it complies with all outstanding technical requirements.

With kind regards,

Yoshikazu Ohya, PhD

Academic Editor

PLOS ONE

Additional Editor Comments (optional):

Reviewers' comments:

Reviewer's Responses to Questions

**Comments to the Author**

1. If the authors have adequately addressed your comments raised in a previous round of review and you feel that this manuscript is now acceptable for publication, you may indicate that here to bypass the “Comments to the Author” section, enter your conflict of interest statement in the “Confidential to Editor” section, and submit your "Accept" recommendation.

Reviewer #1: All comments have been addressed

Reviewer #3: All comments have been addressed

2. Is the manuscript technically sound, and do the data support the conclusions?

Reviewer #1: Yes

Reviewer #3: Yes

3. Has the statistical analysis been performed appropriately and rigorously? 

Reviewer #1: Yes

Reviewer #3: Yes

4. Have the authors made all data underlying the findings in their manuscript fully available?

Reviewer #1: No

Reviewer #3: Yes

5. Is the manuscript presented in an intelligible fashion and written in standard English?

Reviewer #1: Yes

Reviewer #3: Yes

6. Review Comments to the Author

Reviewer #1: (No Response)

Reviewer #3: The authors have adequately addressed reviewer's comments, and the manuscript is now acceptable for publication.

7. PLOS authors have the option to publish the peer review history of their article (what does this mean?). If published, this will include your full peer review and any attached files.

Reviewer #1: No

Reviewer #3: No

---

## [Editor Report · Acceptance letter]

12 May 2020

PONE-D-19-31984R1 

Investigations of the mechanisms of interactions between four non-conventional species with Saccharomyces cerevisiae in oenological conditions. 

Dear Dr. Nidelet:

I am pleased to inform you that your manuscript has been deemed suitable for publication in PLOS ONE. Congratulations! Your manuscript is now with our production department. 

With kind regards,

on behalf of

Dr. Yoshikazu Ohya 

Academic Editor

PLOS ONE